# Number of simultaneously acting global change factors affects composition, diversity and productivity of grassland plant communities

Benedikt Speißer [1] ✉, Rutger A. Wilschut [1,2] & Mark van Kleunen [1,3]

Plant communities experience impacts of increasing numbers of global change factors (e.g., warming, eutrophication, pollution). Consequently, unpredictable global change effects could arise. However, information about multi-factor effects on plant communities is scarce. To test plant-community responses to multiple global change factors (GCFs), we subjected sown and transplanted-seedling communities to increasing numbers (0, 1, 2, 4, 6) of co-acting GCFs, and assessed effects of individual factors and increasing numbers of GCFs on community composition and productivity. GCF number reduced species diversity and evenness of both community types, whereas none of the individual factors alone affected these measures. In contrast, GCF number positively affected the productivity of the transplanted-seedling community. Our findings show that simultaneously acting GCFs can affect plant communities in ways differing from those expected from single factor effects, which may be due to biological effects, sampling effects, or both. Consequently, exploring the multifactorial nature of global change is crucial to better understand ecological impacts of global change.

Global environmental change, mainly caused by humans[1], includes multiple factors[2,3], each with potentially far-reaching ecological consequences[4–7] and socio-economic implications[8]. It is well established that individual global change factors (GCFs) can affect plant-community composition and productivity. For example, increased temperature, eutrophication, plastic, and light pollution can reduce species diversity by promoting the already common species over the less common ones [e.g.,[9–13]]. Furthermore, while light pollution and eutrophication usually affect plant biomass positively[14–16], plastic pollution and salinization usually reduce it[17–20]. In fact, GCFs often differ considerably in their physical, biological, and chemical nature, and thus they can also differ in their ecological effects[21].

In nature, many of these GCFs act simultaneously[21–24]. Yet, there is a lack of information about the joint effects of multiple simultaneously acting GCFs on ecosystem functioning[22,25,26]. Co-acting GCFs may or may not alter the individual effects of other factors. Therefore, the joint net effects of multiple GCFs depend on whether and how these factors interact. As such, the factor-effect relationship could be determined by additive, synergistic or antagonistic effects[27–30]. Consequently, the net effect of co-acting factors might deviate considerably from predictions based on single-factor effects[26,31].

Plant communities could be affected by simultaneously acting GCFs in different ways. Firstly, because many GCFs persistently alter resource availability[32], co-acting GCFs could have profound impacts on niche

[1]Ecology, Department of Biology, University of Konstanz, 78464 Konstanz, Germany. [2]Department of Nematology, Wageningen University and Research, 6708 PB Wageningen, The Netherlands. [3]Zhejiang Provincial Key Laboratory of Plant Evolutionary Ecology and Conservation, Taizhou University, Taizhou 318000, China. ✉e-mail: benedikt.speisser@uni-konstanz.de

dimensionality[33], as this is inversely related to resource availability[34]. As such, the presence of several GCFs that increase resource availability (e.g., eutrophication, light pollution) could synergistically reduce niche dimensionality and consequently species diversity, while GCFs that reduce resource availability (e.g., drought, salinization) could act antagonistically by adding niche dimensions. In fact, a recent synthesis of field studies that manipulated GCFs found that plant-community responses, such as changes in community composition, were more likely when at least three factors were manipulated, indicating interactive effects of co-acting factors[35]. Additionally, combinations of multiple GCFs could also affect ecosystems by causing or shifting tipping points. For example, Polst et al.[36] found that warming reinforced the synergistic interaction of pesticides and nutrient addition, inducing regime shifts in an aquatic system by lowering critical thresholds. Hence, changes in niche dimensionality and the potential to shift tipping points could represent potential pathways of how multiple GCFs interactively impact ecosystems.

Despite the knowledge about potential synergistic and antagonistic interactions and the prospective emergence of novel ecological impacts (e.g., changes in niche dimensionality, tipping points), only few studies have tested the effects of three or more GCFs[25]. A major constraint impeding the investigation of interactive effects of GCFs, is the exponentially increasing number of combinations when more GCFs are experimentally manipulated. For example, a six-factorial experiment would require at least $2^6 = 64$ treatment combinations, which in most cases is logistically not feasible. Moreover, the interpretation of significant four- and higher-order interactions is complex. Recently, however, Rillig et al.[25] presented an elegant approach that allows to examine the effects of increasing numbers of GCFs, but avoids infinite combinations. The approach assumes that in addition to the specific combination of factors, the number of simultaneously acting factors itself may affect biological systems. Effects depending on the number of co-acting GCFs are likely since adding more factors increases the likelihood of including factors of large effect (i.e., a sampling effect) as well as interacting factors[25]. Recent applications of this approach have shown that increasing GCF numbers can negatively affect soil functions and microbial diversity[25], and survival and growth of *Arabidopsis thaliana*[26]. However, studies systematically investigating plant-community responses to increasing co-acting GCF numbers are still missing.

Plant responses to GCFs may depend on the life stage during which plants become exposed. While seed germination and seedling establishment play pivotal roles in the assembly of plant communities, e.g., by determining ecological niches of species[37], plants might be particularly strongly affected by individual or multiple GCFs during these stages. For example, Lloret et al.[38] found that warming reduces species richness among seedlings and that drought reduces the number of seedlings in plant communities. More generally, early plant development can be expected to be especially prone to GCF impacts due to the rapid succession of three critical lifecycle stages (i.e., germination, seedling emergence, and establishment), all of which are sensitive to environmental variables including GCFs[39]. Yet, although global change impacts plant communities at all developmental stages, the early stages have received only minor attention[40,41].

Here, we performed a mesocosm pot experiment to investigate how grassland-plant communities are affected by increasing numbers of concurrently acting GCFs. We exposed communities of nine common herbaceous species that were either started from seedlings (transplanted-seedling communities) or from seeds (sown communities)—thus also including the germination and seedling establishment phases—to zero, two, four, or six simultaneously acting GCFs. We chose six GCFs that differed in their physical and chemical nature, and can frequently act simultaneously: warming, light pollution, microplastic pollution, soil salinization, eutrophication, and fungicide accumulation. We addressed the following questions: (1) Do higher GCF numbers affect the productivity of plant communities? (2) Do higher GCF numbers affect the composition and diversity of plant communities? (3) Are sown communities—due to the inclusion of critical early life stages—especially sensitive to increasing GCF numbers?

## Results

### Productivity of the communities

For both the transplanted-seedling and sown communities, above-ground biomass production was significantly positively affected by eutrophication and negatively affected by microplastic pollution and soil salinization (Fig. 1; Supplementary Tables 1 and 2). While biomass production of the sown community was not significantly affected by increasing GCF number (Fig. 1b; Supplementary Table 3), biomass production of the transplanted-seedling community increased with the number of simultaneously acting GCFs (Fig. 1a; Supplementary Table 4). Subsequent hierarchical diversity-interaction modeling for the transplanted-seedling community showed that the model including separate-pairwise GCF interactions had the best fit (Table 1). This suggests that the positive effect of GCF number on community biomass is due to the identities of the GCFs as well as their specific pairwise interactions.

### Composition and diversity of the communities

In communities exposed to single-GCFs, the first two axes of PCAs on species proportions in the transplanted-seedling and sown communities explained $49.8 + 26.0 = 75.8\%$ and $38.0 + 29.3 = 67.3\%$ of the variation in species composition, respectively (Supplementary Fig. 1a, b). In transplanted-seedling and sown communities exposed to increasing GCF numbers, the first two PC axes explained $41.9 + 26.2 = 68.1\%$ and $41.4 + 24.2 = 65.6\%$ of the variation in species composition, respectively (Supplementary Fig. 1c, d). For the transplanted-seedling community, there were no significant single GCF or GCF-number effects on PC1, but PC2 was significantly affected by GCF number (Fig. 2a; Supplementary Table 4). For the sown community, the PC1 values were significantly altered by microplastics, soil salinization, and warming, and the PC2 values by eutrophication (Fig. 2b, d; Supplementary Table 2). Although PC1 of the sown community was not significantly affected by GCF number, PC2 was significantly affected (Fig. 2d; Supplementary Table 3). GCF-number effects on PC2 were best explained by the separate-pairwise GCF-interactions model for the transplanted-seedling community, and by the GCF-identity model for the sown community (Table 2), suggesting that individual GCFs as well as their specific pairwise interactions contribute to GCF-number effects on species composition.

In both community types, none of the individual single-GCFs affected species diversity or evenness. In contrast, increasing GCF numbers significantly reduced diversity and evenness for both community types (Fig. 3; Supplementary Tables 3 and 4). Subsequent hierarchical diversity-interaction modeling showed that, for the species diversity and evenness of the transplanted-seedling community, the separate-pairwise GCF-interactions model had the best fit. For species diversity of the sown community the GCF-identity model had the best fit. On the other hand, for the species evenness of the sown community, adding GCF identities or GCF interactions did not improve the model, so the null model had the best fit (Table 2).

## Discussion

We investigated the effects of increasing numbers of simultaneously acting GCFs on the productivity, diversity, and composition of plant communities. To assess whether GCF number affects communities, particularly during the early lifecycle stages (i.e., germination and seedling establishment), we started the communities either from seeds or by planting seedlings. The single-GCFs differed in their individual effects and were dependent on the respective community variable (e.g., productivity, diversity; Figs. 1, 3; Supplementary Tables 1 and 2),

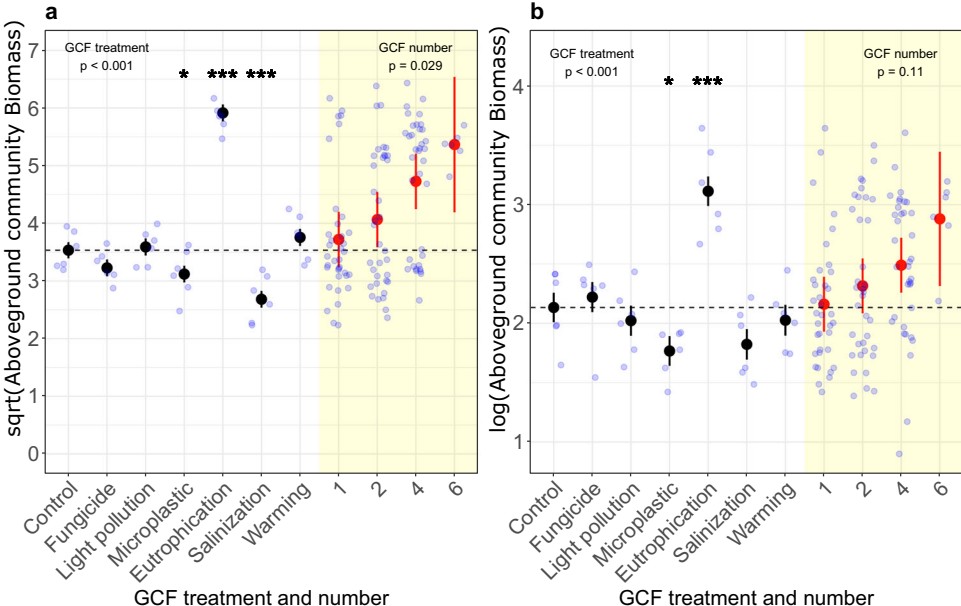

**Fig. 1 | Comparison of single-GCF and GCF-number effects on the productivity of transplanted-seedling and sown communities.** Aboveground biomass of transplanted-seedling (**a**) and sown (**b**) grassland communities exposed to different single-GCF treatments (black circles) and to different numbers of simultaneously acting GCFs (red circles). Asterisks indicate significant single-GCFs (*$p < 0.05$; **$p < 0.01$; ***$p < 0.001$). For the transplanted-seedling community (**a**), the individual GCFs microplastics ($p = 0.048$), eutrophication ($p < 0.001$), and salinization ($p < 0.001$) had significant effects (Supplementary Table 1), and GCF number also had a significant effect (Supplementary Table 4). For the sown community (**b**), the individual GCFs microplastics ($p = 0.026$) and eutrophication ($p < 0.001$) had

significant effects on community productivity, and salinization had a marginally significant effect ($p = 0.065$; Supplementary Table 2). The effect of GCF number was not significant (Supplementary Table 3). Black and red circles represent model estimates for the single-GCF and GCF-number effects, respectively. Error bars represent standard errors. Light blue circles represent raw data distribution. $N = 6$ for all single-GCF treatments, control, and GCF-number level 6, respectively. $N = 36$ for GCF-number levels 1, 2, 4, respectively. General GCF-treatment and GCF-number effects were assessed with log-likelihood ratios ($\chi^2$ statistics). Effects of the individual GCFs were assessed by two-sided t-tests. We did not make adjustments for multiple comparisons.

but their effects were largely consistent between both community types (Supplementary Table 5). The productivity of the transplanted-seedling community increased with GCF number, an effect which was mainly driven by the strong positive effect of eutrophication (as shown by additional analyses; Supplementary Table 6), indicating a strong sampling effect. Yet, the hierarchical diversity-interaction models suggest that in addition to GCF identity (e.g., the strong eutrophication effect), also GCF interactions contributed to the observed positive GCF-number effect on productivity (Table 1). Whereas species diversity and evenness were not affected by any of the single-GCFs, they

## Table 1 | Contributions of GCF identities and GCF interactions to GCF-number effects on the aboveground biomass of the transplanted-seedling communities

| Model (reference model number) | Df | AIC | logLik | $\chi^2$ | $p^*$ |
|---|---|---|---|---|---|
| 1 Null | - | 183.54 | −86.77 | - | - |
| 2 GCF identity (1) | 5 | 162.92 | −71.46 | 30.62 (+) | **<0.001** |
| 3 Separate-pairwise GCF interactions (4) | 11 | 144.28 | −51.14 | 40.64 (+) | **<0.001** |
| 4 Average GCF interaction (2) | 1 | 164.91 | −71.46 | 0.004 | 0.95 |
| 5 Additive GCF-specific interaction contributions (3) | 5 | 162.19 | −65.10 | 27.92 (−) | **<0.001** |

Results of the hierarchical diversity-interaction model comparisons for aboveground biomass. The numbers and names of the models correspond to those in Fig. 4. For each model, we show its AIC and log-likelihood (logLik). To answer the four questions shown in Fig. 4, we made four model comparisons based on log-likelihood ratio tests. The number of the reference model is shown in brackets after the model of interest. For the model comparisons, we provide the degrees of freedom (Df), the $\chi^2$ test statistic and the p value. The + and − in brackets after the $\chi^2$ value indicate whether the model was better or worse than its reference model.
*P values <0.05 are indicated in bold.

declined in both community types with an increasing number of simultaneously acting GCFs (Supplementary Tables 3 and 4). Again, for the transplanted-seedling community, the negative effect of GCF number on species diversity and evenness were best explained by effects of both GCF identities and separate-pairwise interactions between GCFs. The GCF-number effects on species composition and diversity were independent of eutrophication (as shown by additional analyses; Supplementary Table 6), supporting our conclusion that these effects were driven by the number of co-acting GCF and not by the dominant effect of one of the individual GCFs. Our results indicate that the consequences of global change as a multifactorial process on plant communities are unlikely to be predictable based on the effects of individual factors alone, as overall effects can be shaped by specific interactions between co-acting GCFs.

Most studies investigating ecological effects of global change have focused on single-GCFs, resulting in a lack of information about joint effects of co-acting GCFs[22,25,26]. Recently, it was shown that the number of simultaneously acting GCFs by itself can explain changes in soil properties, such as soil respiration, decomposition rate and water-stable soil aggregates, and soil microbial diversity[25]. In another study, it was shown that increasing GCF number negatively affects the growth and survival of a single species, *Arabidopsis thaliana*, and that these effects may differ among genotypes[26]. Our study adds to these findings, by showing that the number of simultaneously acting GCFs increases productivity, decreases diversity and modifies the composition of experimental grassland communities.

Our finding that there are stronger ecological effects when multiple GCFs act simultaneously instead of individually partly contrasts with the conclusions by Leuzinger et al.[42], who found that effects tend to decrease if systems are exposed to more than one GCF. Their dataset, however, contained only studies that manipulated maximally three GCFs, whereas in our study the strongest effects were observed

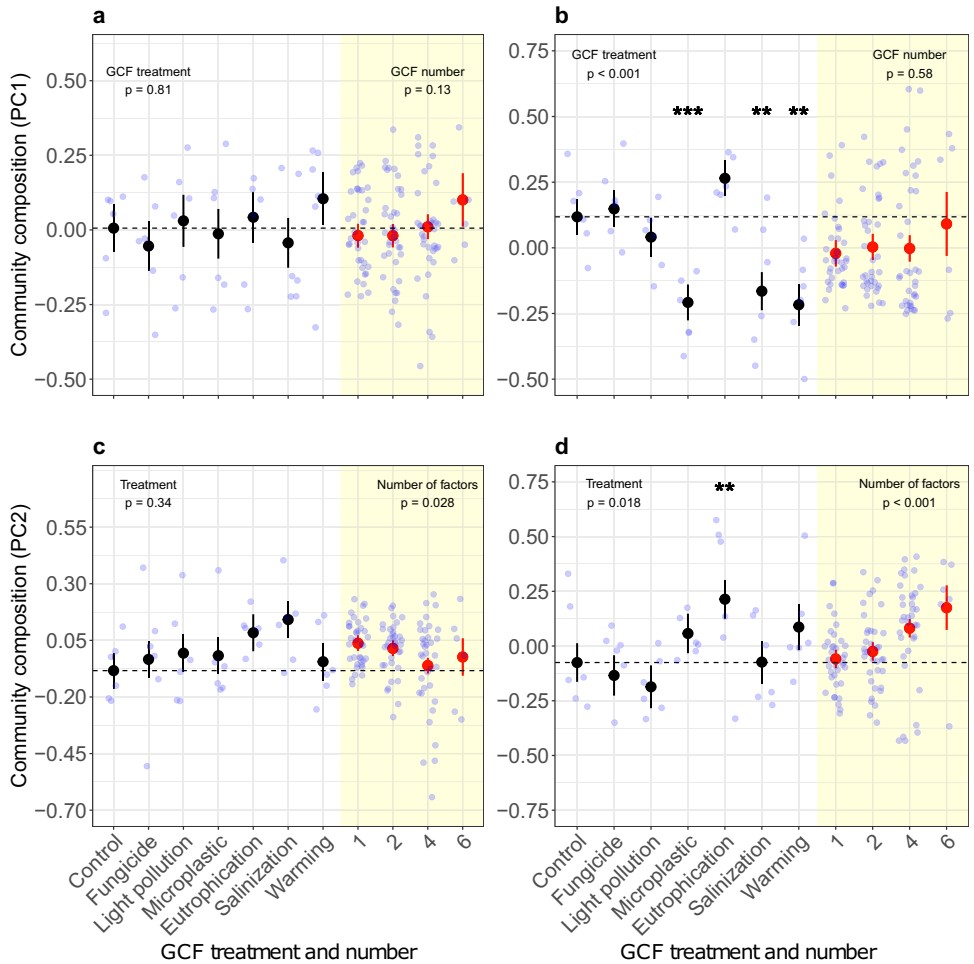

**Fig. 2 | Comparison of single-GCF and GCF-number effects on the species composition of sown and transplanted-seedling communities.** Species composition of transplanted-seedling (**a**, **c**) and sown (**b**, **d**) grassland communities exposed to different single-GCF treatments (black circles) and to different numbers of simultaneously acting GCFs (red circles). Species composition was quantified as the first (PC1; **a**, **b**) and second (PC2; **c**, **d**) axes of a principal component analysis on the proportional biomass of species. In the sown communities, there were significant effects of microplastics ($p < 0.001$), salinization ($p = 0.001$), and warming ($p = 0.001$; PC1; **b**), as well as eutrophication ($p = 0.008$; PC2; **d**) on the community composition (Supplementary Table 2). GCF number had significant

effects on species composition of both community types, based on PC2 (Supplementary Tables 3 and 4). Black and red circles represent model estimates for the single-GCF and GCF-number effects, respectively. Error bars represent standard errors. Light blue circles represent raw data distribution. $N = 6$ for all single-GCF treatments, control and GCF-number level 6, respectively. $N = 36$ for GCF-number levels 1, 2, 4, respectively. Asterisks indicate significance of individual factors ($*p < 0.05$; $**p < 0.01$; $***p < 0.001$). General GCF-treatment and GCF-number effects were assessed with log-likelihood ratios ($\chi^2$ statistics). Effects of the individual GCFs were assessed by two-sided $t$ tests. We did not make adjustments for multiple comparisons.

for higher numbers of co-acting GCFs (Figs.1, 3). Also, the few other studies that explicitly tested for GCF-number effects found the strongest effects when higher numbers of GCFs were combined[25,43]. The latter studies and our results suggest that novel effects can arise when multiple GCFs act simultaneously, potentially leading to "ecological surprises" at high numbers of GCFs sensu, Rillig et al.[25]. The mechanisms underlying the effects of multiple GCFs in our study remain unknown, but likely include direct physiological effects on plant performance, sampling effects, as well as the aforementioned effects on soil properties.

The biomass data we collected in our two-month mesocosm experiment provides clear evidence for short-term effects of increasing GCF number on the productivity, composition and diversity of the plant communities. However, although biomass differences are likely to correlate with differences in survival and reproduction, long-term effects remain hard to predict. A recent synthesis by Komatsu et al.[35] suggests that effects of global change on plant communities increase over time. Furthermore, they found that effects on plant communities were more frequent if three or more GCFs were combined, which is in line with our findings that high GCF numbers can affect plant

communities, even after a relatively short period of time. The short-term GCF effects on the productivity and biomass-based composition of the plant communities could, due to changes in competitive interactions, translate to community shifts in terms of species presence, in the longer run, possibly altering plant-community functioning[44]. However, clear unidirectional community shifts, as observed in response to individual factors [e.g., to drought;[45]], may be less likely when different GCFs are acting simultaneously[46]. For example, while the exposure to an individual GCF will facilitate certain groups of species (e.g., deep-rooted species under dry conditions), trade-offs will likely limit a species' ability to cope with multiple external stressors[47]. Therefore, as global change encompasses a wide range of factors differing in their properties[21], it is unlikely that a large number of plant species can persist in ecosystems exposed to many GCFs. In other words, the more GCFs co-occur, the less likely it gets that a certain species can handle them all.

The overall positive effect of co-acting GCFs on community productivity, although only statistically significant for the transplanted-seedling communities, coincided with a negative effect on species diversity and evenness in both communities. This suggests that

**Table 2 | Contributions of GCF identities and GCF interactions to GCF-number effects on the species composition of both community types**

**Transplanted-seedling community**

| Model (reference model number) | DF | Species diversity (Shannon index) | | | | Species evenness | | | | Species composition (PC2) | | | |
|---|---|---|---|---|---|---|---|---|---|---|---|---|---|
| | | AIC | logLik | $\chi^2$ | p | AIC | logLik | $\chi^2$ | p | AIC | logLik | $\chi^2$ | p |
| 1 Null | - | −69.9 | 39.95 | - | - | −266.15 | 138.08 | - | - | −101.07 | 53.54 | - | - |
| 2 GCF identity (1) | 5 | −63.63 | 41.81 | 3.73 | 0.59 | −260.28 | 140.14 | 4.12 | 0.53 | −101.86 | 58.93 | 10.79 | 0.056* |
| 3 Separate-pairwise GCF interactions (4) | 10 | −80.97 | 61.48 | 39.32 (+) | **<0.001*** | −274.30 | 158.15 | 35.9 (+) | **<0.001** | −99.92 | 68.96 | 19.84 (+) | **0.03** |
| 4 Average GCF interactions (2) | 1 | −61.65 | 41.83 | 0.02 | 0.89 | −258.31 | 140.15 | 0.03 | 0.86 | −100.08 | 59.04 | 0.21 | 0.64 |
| 5 Additive GCF-specific interaction contributions (3) | 5 | −72.65 | 52.33 | 18.31 (−) | **0.005** | −267.08 | 149.54 | 17.22 (−) | **0.001** | −105.80 | 66.90 | 4.12 | 0.53 |

**Sown community**

| Model (reference model number) | DF | Species diversity (Shannon index) | | | | Species evenness | | | | Species composition (PC2) | | | |
|---|---|---|---|---|---|---|---|---|---|---|---|---|---|
| | | AIC | logLik | $\chi^2$ | p | AIC | logLik | $\chi^2$ | p | AIC | logLik | $\chi^2$ | p |
| 1 Null | - | 37.32 | −15.66 | - | - | −105.16 | 55.58 | - | - | −46.83 | 26.417 | - | - |
| 2 GCF identity (1) | 5 | 33.07 | −8.53 | 14.25 (+) | **0.01** | −102.80 | 59.40 | 7.64 | 0.18 | −53.01 | 34.50 | 16.17 (+) | **0.006** |
| 3 Separate-pairwise GCF interactions (4) | 10 | 41.65 | −1.83 | 13.40 | 0.20 | −85.83 | 61.91 | 4.45 | 0.92 | −40.16 | 39.08 | 9.07 | 0.53 |
| 4 Average GCF interactions (2) | 1 | 35.06 | −8.53 | 0.01 | 0.91 | −101.38 | 59.69 | 0.57 | 0.45 | −51.09 | 34.54 | 0.08 | 0.77 |
| 5 Additive GCF-specific interaction contributions (3) | 5 | 35.17 | −3.58 | 3.51 | 0.62 | −94.30 | 61.15 | 1.52 | 0.91 | −49.06 | 38.53 | 1.10 | 0.95 |

Results of the hierarchical diversity-interaction model comparisons for community composition (PC2), diversity, and evenness. The numbers and names of the models correspond to those in (Fig. 4). For each model, we show its AIC and log-likelihood (logLik). To answer the four questions shown in Fig. 4, we made four model comparisons based on log-likelihood ratio tests. The number of the reference model is shown in brackets after the model of interest. For the model comparisons, we provide the degrees of freedom (Df), the $\chi^2$ test statistic and the p value. The + and − in brackets after the $\chi^2$ value indicate whether the model was better or worse than its reference model.
*P values <0.05 are indicated in bold, and p values <0.1 are indicated in italics.

productivity and species structure of plant communities respond differently to the exposure to multiple factors[33], as the increase in productivity is not equally accounted for by the different species. The results of the diversity-interaction modeling suggest that the GCF-number effects on species diversity, which mainly reflect changes in species evenness, and on community productivity could be best explained by models that included both the GCF identities and specific pairwise GCF interactions (Tables 1, 2). However, for the species diversity and evenness of the transplanted-seedling community, a model just including the GCF identities was not better than the null model, suggesting that GCF interactions were especially important. The latter is in line with our finding that none of the six single-GCFs individually affected the diversity of the transplanted-seedling communities. For diversity of the sown community, however, including GCF identities alone already resulted in a better model than the null model. Our finding that the combination of multiple GCFs can reduce species diversity and evenness of plant communities despite the absence of any effect of the single-GCFs shows that ecological responses to multiple GCFs cannot always be predicted from the responses to single factors, but are rather shaped by complex interaction effects[27].

As the transplanted-seedling communities were harvested two weeks earlier than the sown communities, we analyzed them as separate experiments. Nevertheless, to test explicitly for differences between them, we also did additional joint analyses, which showed that the transplanted-seedling communities produced more biomass and had a higher species diversity than the sown communities (Supplementary Fig. 2, Supplementary Table 5). As the transplanted-seedling and sown communities did not differ in evenness, the higher diversity of the transplanted-seedling communities most likely reflects that the transplanted seedlings of most species survived, whereas in the sown communities some species where poorly represented (Supplementary Figs. 3 and 4). In particular, the grass *Poa pratensis* was absent from almost all of the sown communities (Supplementary Fig. 3), most likely because it had failed to germinate. Further, GCF number had a generally negative effect on the number of plants in both community types, i.e., reduced the survival of plants and, for the sown community, probably also the germination success (Supplementary Fig. 3, Supplementary Table 7). The joint analyses showed that the effects of the single-GCFs on biomass production and diversity were largely the same for the transplanted-seedling and sown communities. Also, the effects of GCF number were largely the same for both community types, although the negative effect of GCF number on species diversity and evenness were slightly stronger for the sown community than for the transplanted-seedling community (Fig. 3; significant and marginally significant GCF number × Community interactions in Supplementary Table 5). This again most likely reflects that, in the sown community, diversity and evenness were not only affected by differences in survival and growth but also by differences in germination. This is in line with previous findings that global change may have particularly strong effects on plant performance during the germination phase[39,48], suggesting that plant communities might be especially susceptible to global change due to effects on seed germination[41,49].

The increased productivity of transplanted-seedling communities and the reduced species diversity and evenness of both sown and transplanted-seedling communities show that higher numbers of co-acting GCFs can have profound effects on plant communities. These effects of co-acting GCFs are partly dependent on the identities of the co-acting GCFs, as indicated by increases in model fit (lower AIC values and higher log-likelihood values) when the GCF-identity models were compared to the null models (significant for biomass production of the transplanted-seedling communities and species composition and diversity of the sown communities; Tables 1 and 2). Yet, while individual effects of single-GCFs on plants can get weakened when combined with more and more other GCFs[47], in contrast, overall effects of

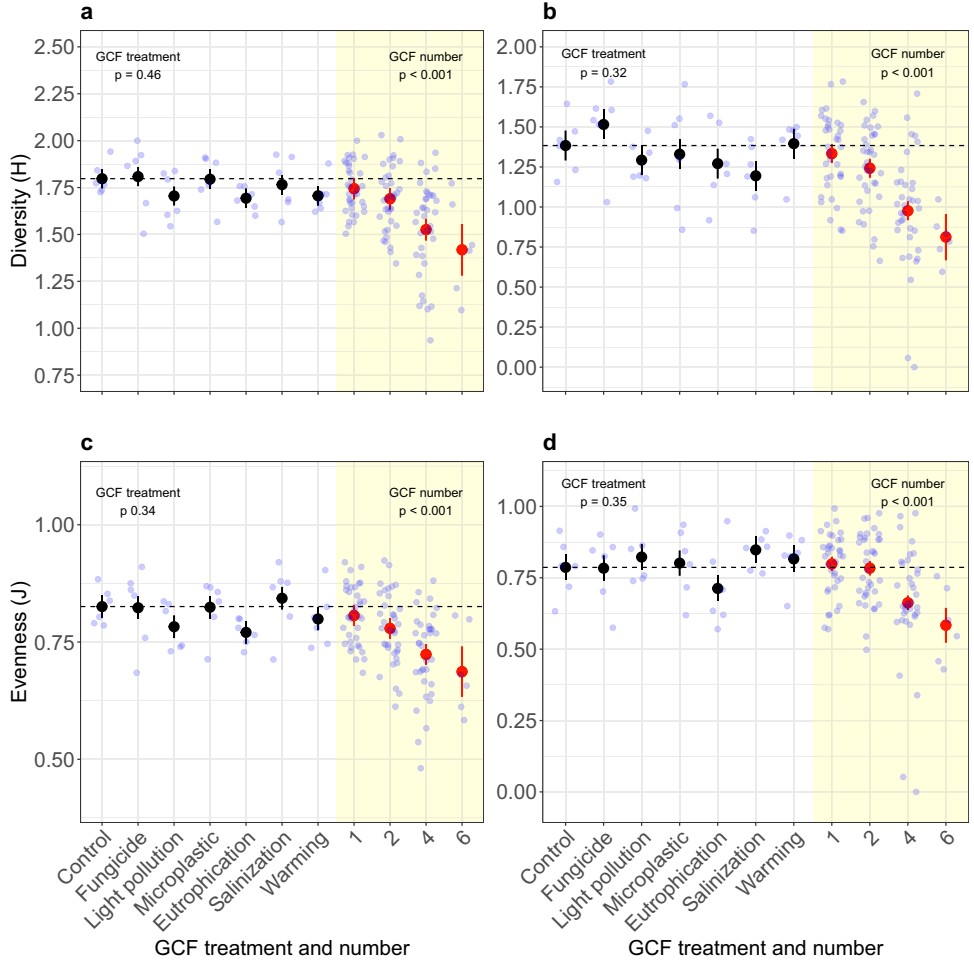

**Fig. 3 | Comparison of single-GCF and GCF-number effects on species diversity of sown and transplanted-seedling communities.** Species diversity and evenness of transplanted-seedling (**a**, **c**) and sown (**b**, **d**) grassland communities exposed to different single-GCF treatments (black circles) and to different numbers of simultaneously acting GCFs (red circles). Species diversity was quantified as Shannon index (**a**, **b**) and as evenness (**c**, **d**) based on the proportional biomass of the species. In both community types, both species diversity and evenness were significantly reduced by GCF number while there were no effects of the single-GCF treatments (Supplementary Tables 3 and 4). Black and red circles represent model estimates for the single-GCF and GCF-number effects, respectively. Error bars represent standard errors. Light blue circles represent raw data distribution. $N = 6$ for all single-GCF treatments, control and GCF-number level 6, respectively. $N = 36$ for GCF-number levels 1, 2, 4, respectively. Asterisks indicate the significance of individual factors (*$p < 0.05$; **$p < 0.01$; ***$p < 0.001$). General GCF-treatment and GCF-number effects were assessed with log-likelihood ratios ($\chi^2$ or F statistics). Effects of the individual GCFs were assessed by two-sided $t$ tests. We did not make adjustments for multiple comparisons.

simultaneously acting GCFs might be especially pronounced at higher GCF numbers[25,43]. Therefore, to gain insight into general effects of GCF number, future studies should especially focus on higher numbers of combined factors. Furthermore, to better disentangle the individual GCF and GCF-number effects, future studies should include multiple different GCF combinations, also for the highest GCF-number levels. This will require a pool of possible GCFs that exceeds the highest GCF-number level in the experiment. It should also be noted that, although we used realistic levels of the factors (Supplementary Table 8), they may occur at a wide range of different intensity levels in nature[50–52], each with potentially different ecological consequences[9,18,19,53]. Hence, to improve our understanding of how global change can impact biological systems, future studies should not only include more GCFs but also multiple intensity levels of each GCF.

Our study shows that the number of simultaneously acting GCFs can affect the productivity, species composition and diversity of plant communities. The mechanisms behind such effects might differ, as indicated by the potential influences of sampling effects underlying the biomass responses but not the effects on species diversity. Especially the negative impacts of GCF number on species diversity, mainly reflecting a decreased evenness, suggest that global changes may have more devastating effects on terrestrial biodiversity than indicated by studies that focused on single or few GCFs. Furthermore, our findings suggest that early plant lifecycle stages are of high relevance when it comes to evaluating the effects of global change on plant communities, as the sown communities were more susceptible to GCF number than the transplanted-seedling communities. Our findings emphasize the potential implications of global change as a multifaceted process and corroborate the need for more holistic approaches in global change research.

## Methods

### Study species and pre-cultivation

To create the mesocosm communities, we selected nine herbaceous grassland species that are native to and widespread in Central Europe (Supplementary Table 9), where they can also co-occur. The species were *Alopecurus pratensis* L., *Diplotaxis tenuifolia* (L.) DC., *Lolium perenne* L., *Poa pratensis* L., *Prunella vulgaris* L., *Sinapis arvensis* L., *Sonchus oleraceus* L., *Vicia cracca* L., *Vicia sativa* L. To increase generalizability[54], the species were selected from three functional groups (grasses, annual forbs, perennial forbs), and they represent five families.

Seeds were obtained from different sources (Supplementary Table 9). For the transplanted-seedling community (see section 'Experimental lay-out), seedlings were pre-cultivated in a greenhouse of the Botanical Garden of the University of Konstanz. As the species require different times for germination, they were sown on different dates (Supplementary Table 10) to ensure that seedlings of all species were at a similar developmental stage at transplantation. Seeds were sown separately per species in plastic trays filled with potting soil (Einheitserde®, Pikiererde CL P). The greenhouse had a regular day-night rhythm of c. 16:8 hours, and its ventilation windows automatically opened at 21 °C during the day and at 18 °C during the night. Two days before transplanting, the seedlings were placed outdoors to acclimatize. For the sown community, we sowed a seed mixture of the nine study species directly into the outdoor mesocosm pots.

## Experimental setup

**Global change treatments.** We imposed six global change treatments: climate warming, light pollution, microplastic pollution, soil salinization, eutrophication, and fungicide accumulation, all of which frequently occur in the environment. These GCFs were chosen because they differ in their nature (i.e., physical, chemical), are likely to differ in their mode of action and effect direction[21], and can be easily implemented. Each of the six GCFs have been shown to impact plants and their environment when applied on their own[10,13,17,19,20,55–60]. Furthermore, all of the chosen GCFs are likely to continue to increase in magnitude or extent in the near future[61–65]. For the climate-warming treatment, we used infrared-heater lamps (HS-2420; 240 V, 2000 W; Kalglo Electronics Co., Bethlehem, USA) set to 70% of their maximum capacity to achieve an average temperature increase of 2.0 °C (±SD = 0.2 °C) at plant level. This is within the range of temperature increases predicted by the RCP 4.5 scenario for the year 2100 [+1.1 − 2.6 °C; 63]. For the light-pollution treatment, we used LED spotlights (LED-Strahler Flare 10 W, IP 65, 900 lm, cool white 6500 K; REV Ritter GmbH, Mömbris, Germany), which were switched on daily from 9 pm to 5 am, corresponding to the times of sunset and -rise. The average light intensity was 24.5 lx at ground level, which is within the range of light intensities found below street lights, and matches the light intensities used in other light-pollution experiments[14,56]. For the microplastic pollution treatment, we used granules (1.0–2.5 mm diameter) of the synthetic rubber ethylene propylene diene monomer (EPDM Granulat, Gummi Appel GmbH + Co. KG, Kahl am Main, Germany) at a concentration of 1% (w/w, granules/dry soil, approximately corresponding to 1.5% v/v). EPDM granules are, for example, used in artificial sport turfs, from where they easily spread into the surroundings, and have been used previously to investigate the effects of microplastics on plants[18]. The chosen concentration is well within the range of concentrations used in previous studies[18,66,67], and is at the low to intermediate range of concentrations found in sites polluted with plastics[68]. For the soil-salinization treatment, dissolved NaCl was added to the soil. Soil salinity is commonly measured as electrical conductivity, with a conductivity between 4 and 8 dS m$^{-1}$ considered to be moderately saline[69]. For the experiment, we used a salinity of 6 dS m$^{-1}$. To maintain a more or less constant salinity level, electrical conductivity was measured weekly, and, if required, adjusted by adding dissolved NaCl. For the eutrophication treatment, 3 g of a dissolved NPK fertilizer (Universol® blue oxide, ICL SF Germany & Austria, Nordhorn, Germany) was added per pot. For N, this corresponds to an input of 100 kg N ha$^{-1}$, comparable to the yearly amounts of atmospheric N deposition in large parts of Europe[52] and the yearly nitrogen input on agricultural field in the European Union[70]. To ensure a more or less constant nutrient availability during the experiments, we split total fertilizer input into three applications (directly after, 3 weeks after, and 6 weeks after starting the experiments) of 1 g fertilizer per pot per application. In addition, to avoid severe nutrient limitation in the other pots, all pots (irrespective of the eutrophication treatment) received basic fertilization. This was applied four times to the transplanted-seedling-community pots and five times to the sown community pots, with 0.2 g fertilizer per pot per application. For the pesticide treatment, we used the fungicide Landor® CT (Syngenta Agro GmbH, Maintal, Germany). This fungicide was chosen because it contains three azoles as active agents, which belong to the most widely used class of antifungal agents[71]. To each pot in this treatment, we added 1.5 μl fungicide dissolved in water (1‰). This corresponds to 60% of the maximum amount that should be used per hectare of cropland. A summary of the levels of the individual GCFs used in our experiment is provided in Supplementary Table 8.

**Combinations of simultaneously acting GCFs.** To examine the potential effects of the numbers of simultaneously acting GCFs, we created five levels of increasing GCF numbers. These levels were: zero (i.e., the control without any GCF application), one (single), two, four and six GCFs. For the one-, two- and four-GCF levels, there were six different combinations, so that each of these levels included either six different GCFs in case of the one-factor, or six different GCF combinations in case of the two- and four-GCF levels. In the six-GCF level, all six factors were combined, so that there was only one combination. To avoid potential biases due to unequal representation of the different GCFs in each GCF-number level, we created the GCF combinations randomly but with the restriction that each GCF was present in an equal number of combinations for each GCF-number level (i.e., each GCF was included once in GCF-number levels 1 and 6, respectively, twice in GCF-number level 2, and four times in GCF-number level 4; Supplementary Table 11).

**Experimental lay-out.** The experiment was conducted outdoors in the climate-warming-simulation facility of the Botanical Garden of the University of Konstanz, Germany (N: 47°69′19.56″, E: 9°17′78.42″). Twenty of the 2 m × 2 m plots of this facility were used for our experiment. As the climate-warming and light-pollution treatments could not be applied to each individual pot separately, we applied those treatments at the plot level. Therefore, we assigned four of the 20 plots to the climate-warming treatment, four plots to the light-pollution treatment and four plots to both climate-warming and light-pollution treatment combination. Each plot had a 145 cm high metal frame. The eight plots assigned to the climate-warming treatment were equipped with a 1.80 m long, horizontally hanging infrared-heating lamp at the top of the metal frame (i.e., at 145 cm above soil level). The heating lamp slowly oscillated along its longitudinal axis to ensure uniform heating of the whole 2 m × 2 m plot. The eight plots assigned to the light-pollution treatment, each had a LED spotlight attached to one of the sides of the metal frame at a height of 120 cm. To reduce illumination of the neighboring plots, light-pollution was only applied to the outer plots of the climate-warming-simulation facility (Supplementary Fig. 5), and LEDs were pointing away from the inner plots and were equipped with lamp shades made of black plastic pots (18 cm × 18 cm × 25.5 cm). Furthermore, to reduce the light intensity to a realistic light-pollution level (24.5 lx) as found below street lights, we covered the spotlight with a layer of white cloth (Supplementary Fig. 6). For further details on the artificial light treatment, see Supplementary Fig. 7.

To create mesocosms with the transplanted-seedling and sown communities, we filled 10-L pots (CEP- Container, 10.0 F, Burger GmbH, Renningen-Malmsheim, Germany) with a mixture of 40% potting soil (see above), 40% quartz sand (0.5–0.8 mm), and—to inoculate the substrate with a natural soil community—20% top soil excavated from a seminatural grassland patch in the botanical garden. In total, the experiments with the transplanted-seedling and sown communities, each included 120 pots (i.e., 20 treatment combinations × six replicates × 2 experiments = 240 pots in total; see Supplementary Table 11), which were distributed across the 20 plots. To prevent leakage of fertilizer or salt solutions, each pot was placed onto a plastic

dish. To reduce differences due to environmental variation within plots, the positions of pots within each plot were re-randomized every 14 days. Plants were watered regularly to avoid drought stress and to avoid differences in soil moisture due to application of fertilizer- and salt-solutions.

For the sown community, we randomly distributed five seeds of each of the nine species on the substrate in each pot on 3 July 2020. For the transplanted-seedling community, two seedlings of each of the nine species were transplanted into each pot (i.e., 18 seedlings per pot) according to a fixed pattern (Supplementary Fig. 8) on 6 July 2020. Since there were a few seedlings missing for *S. arvensis* (six seedlings) and *V. cracca* (four seedlings), we re-sowed these species in germination trays on 6 July 2020. On 13 July 2020, dead seedlings, and the missing seedlings for *S. arvensis* were replaced. Since *V. cracca* took longer to germinate, the missing seedlings were transplanted on 17 July 2020.

### Measurements
To investigate the effects of single-GCFs and their number on the sown and transplanted-seedling communities, we used plant biomass as an indicator for plant performance[72]. As it was impossible to disentangle the roots, we only used aboveground biomass. On 14 and 15 September 2020, i.e., 10 weeks after transplanting, we harvested the transplanted-seedling communities. On 28 and 29 September, i.e., twelve weeks after sowing, we harvested the sown communities. For both community types, we harvested the plants separately by species. The harvested plants were stored in paper bags, dried at 70 °C for at least 72 hours and weighed.

### Statistical analysis
All analyses were done in R 3.6.2[73]. As the transplanted-seedling and sown communities were harvested at different times, we treated them as separate experiments, and therefore analyzed them separately (but see the subsection "Community type specific responses" below).

**Community aboveground biomass.** To analyze the effects GCF number on plant-community productivity, we fitted linear mixed-effects models separately for the transplanted-seedling and sown communities, using the *lmer* function in the "lme4" package[74]. Total aboveground biomass per pot was the response variable. To improve normality of the residuals, biomass of the transplanted-seedling and sown communities was square-root- and natural-log-transformed, respectively. We included GCF number as a continuous fixed variable. To account for non-independence of pots in the same GCF combination and of pots in the same plot, GCF combination and plot were included as random effects. The effects of the individual GCFs on biomass production were also assessed by fitting linear mixed-effects models, using only the data of the control and single-GCF treatments, and including GCF identity as fixed effect.

**Community composition.** To assess potential effects of single-GCFs and GCF number on the final composition of the transplanted-seedling and sown communities, we first assessed variation in species composition, based on biomass proportions, among pots using principal component analysis (PCA) [rda function of the "vegan" package[75],]. For each PCA (Supplementary Fig. 1), we extracted the PC1 and PC2 values, which together explained more than 65% of the variation in community composition and included them as response variables in separate linear mixed models, as described above for community biomass.

To evaluate whether GCF number affects the diversity and evenness of plant communities, we calculated the Shannon index (H)[76], using the diversity function in the "vegan" package, and evenness index (J)[77] based on species biomass proportions. Subsequently, the single-GCF and GCF-number effects on diversity and evenness of the sown and transplanted-seedling communities were analyzed using

linear mixed-effects models, or–if adding random effects did not improve the model–more parsimonious linear models[78,79]. For all models, we used type II analysis of variance (ANOVA) tests (Anova function in the "car" package) to assess the significance of fixed effects.

**Hierarchical diversity-interaction modeling.** When there is a significant GCF-number effect, this could reflect that with increasing numbers of co-acting GCFs, there is a higher chance that it will include a GCF with a strong and dominant effect (i.e., sampling or selection effects). However, it could also be that the GCF-number effect is driven by interactions among the GCFs, and the effects of these interactions could be GCF-specific or general. As our experiment does not include all possible combinations of GCFs, it does not allow to test the contributions of each possible multi-way GCF interaction. Therefore, to gain insights into whether the GCF identities and specific or general GCF interactions underlie the significant GCF-number effects, we applied the hierarchical diversity-interaction modeling framework of Kirwan et al.[80]. This framework was originally developed for estimating contributions of species identities and their interactions to ecosystem functions, but we here applied it to GCF identities and interactions. For each of the response variables showing a significant GCF-number effect, we ran five hierarchical models specifying different assumptions about the potential contributions of individual GCFs and their interactions to the GCF-number effect, and compared them using likelihood ratio tests (Fig. 4). For these analyses, the data of the control treatment (i.e., GCF number zero) was excluded. Each of the five models specified different assumptions about the potential contributions of individual GCFs and their interactions to the GCF-number effect. The first model is the null model, which assumed that there were no GCF-specific contributions (i.e., all GCFs contributed equally) and that there were no contributions of GCF interactions. Therefore, the null model only included the centered sum of the GCFs of each treatment (M) as fixed effect. M accounts for differences in 'initial abundances' of GCFs– meaning that the other model terms are interpreted based on the average initial abundance–and was also included in the four other models[80]. This way, we could include the GCFs' relative proportions in each GCF combination, instead of just considering GCF presence, while taking into account that, with increasing GCF number, the relative proportion of each individual GCF is automatically reduced. In the second model, the GCF identities (i.e., their proportions in the respective GCF combination) were added, assuming that individual GCFs contribute differently to the effect of GCF number. In the third model, separate-pairwise interactions between the GCFs were added, considering that, in addition to contributions of individual GCFs, specific pairwise interactions contributed to the GCF-number effect. In the fourth model, the average GCF-interaction model (which is also called the evenness model in Kirwan et al. 2009), the separate-pairwise GCF interactions were replaced by an average interaction effect. Thus, the average GCF-interaction model assumed equivalent contributions of all pairwise GCF interactions. In the fifth model, the additive GCF-specific interaction contributions model, the average interaction effect of the fourth model was replaced by average GCF-specific interaction effects. This model assumed that each GCF's contribution to a pairwise interaction remains constant. For the calculation of the average GCF-specific and average interaction effect, we used the equations provided by Kirwan et al.[80]. For each of the response variables, we generally included the same random terms as in the main analyses of the GCF-number effect. However, as this resulted in singularity warnings for some of the hierarchical diversity-interaction models, e.g., those for species diversity and evenness measures, we used for these cases linear models instead of linear mixed models.

**Community type-specific responses.** As the transplanted-seedling and sown communities were harvested at different times, we treated them as separate experiments, and therefore analyzed them separately.

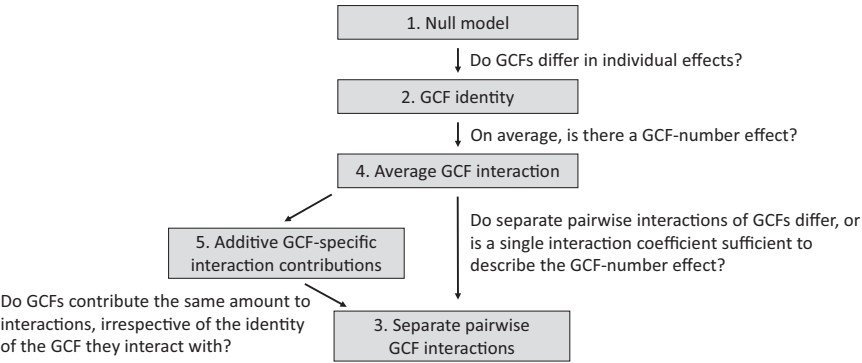

**Fig. 4 | Hierarchical diversity-interaction-modeling framework to assess contributions of GCF identities and GCF interactions to GCF-number effects.** The framework was adapted from Kirwan et al.[80]. The null model assumes equivalent contributions of all GCFs and no interactions between them. The subsequent models assume more complex effects of how the individual GCFs and their interactions determine the GCF-number effects. The questions that can be answered by comparing specific models are depicted next to the arrows connecting the two models.

However, to test explicitly whether both community types differed in their responses to single-GCFs and GCF number, we also analyzed them jointly. To this end, we fitted linear mixed-effects models for each response variable including GCF number (or single-factor treatments), community type and their interaction as fixed effects (Supplementary Table 5).

**Final number of plants per species.** To test for effects of individual GCFs and GCF number on species presence, i.e., the number of individuals per species present at harvest, we fitted generalized linear mixed-effects models for the transplanted-seedling and sown communities separately. We included the survival rate (number of individuals present at harvest divided by the number of planted/sown individuals) as response variables. For the models testing the effects of GCF number, we included GCF combination, species, pot, and plot as random effects. For the models testing the effects of single-GCFs, the same random effects were included, except for GCF combination. Specific random effects were removed from the model if their incorporation resulted in singular fit warnings due to low variation. We assessed the effects of individual GCFs or GCF number using type III ANOVA tests (Anova function in the "car" package, Supplementary Table 7).

**Eutrophication effects.** In addition to the general assessment of individual GCF effects in the hierarchical diversity-interaction models, we specifically assessed the effects of eutrophication. This was done because eutrophication had the strongest effect on productivity as individual GCF, and this might also have dominated the GCF-number effect, indicating a sampling effect. To this end, we added a binary-coded variable to include information on whether eutrophication was included in the different GCF combinations. Subsequently, we fitted linear mixed-effects models for all response traits that were affected by GCF number. In these models, we included GCF number, community type, eutrophication, and the respective two-way interactions as fixed effects, and plot and GCF combination as random effects. Effects of fixed factors were assessed using type III ANOVA tests (Anova function in the "car" package; Supplementary Table 6).

### Reporting summary
Further information on research design is available in the Nature Portfolio Reporting Summary linked to this article.

## Data availability
All data used for the analyses were obtained from the experiment. All data are available at https://doi.org/10.6084/m9.figshare.21625292.v1.

## Code availability
The R code for the statistical analyses and the figures is available at https://doi.org/10.6084/m9.figshare.21625292.v1.

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

## Acknowledgements

We thank Otmar Ficht, Heinz Vahlenkamp, Beate Rüter, and Gretchen Garcia for practical assistance, and Zhijie Zhang, Marc Stift, and Trevor Fristoe for their input regarding the statistical analyses.

## Author contributions
M.v.K and R.A.W. conceived the idea. M.v.K., R.A.W., and B.S. designed the experiment. B.S. performed the experiment and analyzed the data with additional inputs by M.v.K. and R.A.W. B.S. wrote the first draft of the manuscript with further contributions by M.v.K. and R.A.W.

## Funding

## Competing interests
The authors declare no competing interests.
