## [Peer Review File · Nature Communications]

Reviewer comments, first round -

Reviewer #1 (Remarks to the Author):

The manuscript presented by Speisser et al presents the results of a pot experiment that investigates the effects of multiple simultaneous global change drivers on plant community composition and productivity. The study builds upon work by Rilling et al to examine not the exact combinations of global change factors manipulated, but rather the number of factors (factor richness). On the whole, I appreciate this approach to addressing the important question of consequences of multiple interacting global change factors. However, I have a few major comments related to the structure of the manuscript and statistical analysis.

While it is valid that the number of global change factors examined here cannot be factorially manipulated, I would suggest that this not form the majority of the set-up of the introduction. Rather, it would be interesting to consider why more simultaneous factors might impact plant communities and growth from an ecological theory standpoint. For example, does it reduce the number of niches available or lead to tipping points? Incorporating a theoretical basis for why the richness of global change factors would matter would be more compelling than the current methods-focused rationale for the experimental design.

Similarly, the discussion could benefit from incorporation of more theory. As is, the literature cited is mainly used to either say "they did find the same pattern" or "they didn't find the same pattern", with no discussion of why those differences might have been observed.

The experimental design to assess community composition needs further clarification. It is unclear why a mixed effects model using scores from the first PCA axis is used to assess community composition, when other statistical techniques are more suited for community analysis. For example, a PERMANOVA (Anderson, 2001, *Austral Ecology*, 26: 32-46) would be a better choice to assess changes in community composition in the pots and can also be performed using the vegan package in R.

Further, Shannon's H index incorporates both evenness and richness into the calculation. Were plants allowed to die in the planted communities in a way that would impact richness? Or was this metric primarily used to assess differential species richness in the sown (seeded) pots? And if so, is there any measurement of germination success in the sown pots to differentiate death vs germination as the source of variation in the response? I would suggest that evenness alone, instead of Shannon's H be used for the planted pots if death was not observed during the course of the experiment (which I think likely given the short-term nature of the study). In the discussion it is implied that the cause was differential germination, but evidence of this should be provided.

If assessing the consequences of communities growing from seed vs planted plugs is a primary goal of this study, I would suggest that this should be incorporated into the mixed models as a fixed factor. Otherwise comparisons between the two are difficult to support given the current stats.

Clarification throughout of the effects of global change factor richness on the response variables is needed. In the first paragraph of the discussion, the authors suggest that in the planted pots the interaction of multiple factors combined in unique ways that cannot be predicted by individual factors alone when describing the plant community (Ins 277-281), but for the productivity response they suggest that the effects of factor richness might have been driven by the positive effects of eutrophication (i.e., a selection effect) (Ins 274-275). Yet the results in Table 1 show for both productivity and community responses, the best model of interaction was "separate-pairwise interactions between factors". If that were the case, then the same conclusions should be drawn for both community and productivity responses, not the alternative conclusions listed above. However, in Ins 336-340 it is stated that the "factor identity model" was the best fit until the separate-pairwise interactions was added to the model. Together the discussion of these factor richness drivers is confusing throughout and care should be taken to unify the discussion text with

the statistical results.

Additional minor comments:

(1) Somewhere in the title, abstract, and/or keywords, it should be made clear what kind of plant species this experiment was conducted on (i.e., herbaceous species).

(2) Some relevant literature is missing from the paper, including the following:

a. Leuzinger et al, 2011, Trends in Ecology and Evolution, 26: 236-241

b. Langley et al, 2014, AoB Plants 6: 1-12

(3) The distinction between sown and planted communities should be made more clear in the intro/abstract. Different terms might be better to make the distinction (e.g., plugs vs seeded communities).

(4) The entire paragraph on soil biophysical properties in the discussion (lns 323-331) is very speculative given that soils were not measured in this experiment. I would suggest removing this paragraph entirely.

Reviewer #2 (Remarks to the Author):

Review of Speiber

The authors conducted a mesocosm study to test the effects of multiple factors on plant biomass and community diversity. The focus is on how the number of global change factors affects plant communities. The authors conclude that multifactorial combinations yield effects that are not predictable from single factor effects. The topic of how multiple factors interact is of great interest. There have been many individual studies and several reviews focused on this very question with equivocal conclusions.

The presentation of the data here do not strongly support the conclusions. The simplest explanation for the most notable pattern is that eutrophication, by far the most dominant single treatment, caused the higher biomass in the treatments with higher treatment richness. The team used hierarchical diversity models to demonstrate that models with separate pairwise interactions fit the data better than alternatives, indicating that those were better than models with factor identity. I feel like this is a roundabout and opaque way to communicate these data. I would be more satisfied by maybe analyzing a subset of the data with eutrophication removed. That seems like a more straightforward way to eliminate the effects of this dominant treatment. At the very least, there should be some graphical representation of the results that support this primary finding. Can we see the strength or the spread of these two-way interactions? Additionally, perhaps the average absolute value of treatment effects could be shown instead of means across treatments and treatment combinations that may go in opposite directions and cancel each other out. I need some stronger convincing that eutrophication had no effect on diversity or richness, and that unique combinations of treatments are driving the patterns.

I'm not sure that PC1 is the best way to capture community differences. I feel like there could be a lot of important info missing in other dimensions.

Intro: No rationale is given for hypothesis 3.
Why were these particular treatments chosen?

Line 108 Do these species co-occur naturally?

L137 The supplement says the eutrophication rate was 100 kg ha⁻¹ of Universol® blue (oxide) fertilizer, which included NPK and micronutrients, while the methods say that the eutrophication treatment is 100 kg ha⁻¹ N. NPK would be expected to have much more profound effects than just N bc of nutrient colimitation.

L184. Only 2 months may be especially short for diversity patterns to manifest.

L332. It seems like factor richness has the same effects as we would expect for fertilization: more biomass, less diversity, less evenness. This effect has been shown in many hundreds of studies. I would be surprised if it were not the same here.

L340 what is the factor-dominance mechanism? Has that been introduced?

L349 were there any marginally significant effects? I would guess that eutrophication must have been close. Perhaps, just by chance it did not meet the criteria for significance alone, but in combination with other

I would expect that some of these factors are very trivial for some species (maybe fungicide has very little influence on a species that relies weakly on mycorrhizal fungi), but eutrophication may have a strong effect on all species.

L282. That these factors have different effects on plants

L670 "assess"

Throughout: avoid ambiguous antecedents, starting a sentence with "This [verb]...". It often forces the reader to go backwards before continuing with the sentence.

Fig. 2 and 3 are both sandwiched such that the data are difficult to see. If the y-axis were expanded, it would be easier to see the differences between the points and to see the error bars. Also, it would be helpful to see the individual datapoints in each group- it would convey a feel for the replication each mean and error represents.

Reviewer #3 (Remarks to the Author):

This manuscript by SpeiBer et al. entitled "Multiple simultaneously acting global change factors change plant-community composition and productivity" describes a really lovely test of the hypothesis that simultaneous global change factors may result in different changes to ecosystems than individual global change factors. This paper represents the first effort that I am aware of to test this hypothesis using the number of concurrent global change factors as the main variable in question.

In general – I love this paper. I think it is clearly written and easy to understand and provides great evidence for the importance of thinking about multiple global change factors simultaneously despite the logistical difficulties of doing so. I find the results to be compelling. I also appreciate that the authors take in pieces of the Biodiversity-Ecosystem Functioning framework and use them in an entirely new way that is both unexpected and absolutely spot on. Below – I highlight a few ways in which I think that this already great manuscript can be improved.

1. In general, while I found the manuscript to be very clear and largely in active voice. I found the use of lots of specific terms to be quite jargony. As someone in the BEF field – these terms don't necessarily make it harder for ME to understand the paper but I am concerned that they will decrease the impact of the message of this paper to people outside of this relatively narrow field. Considering the diverse readership of Nature Communications, I would suggest that the authors exchange these terms for phrases that better describe their meaning. Some examples below (though this is a non-exhaustive list just examples that stuck out).

a. Factor richness – I appreciate this analog to species richness but I think it obscures what the variable really means which is just the number of simultaneous global change factors applied. I would use this more intuitive phrase to make it more clear to the readers.

b. Planted vs. sown communities – I think planted vs. sown is a good distinction for plant ecologists but won't evoke an obvious difference to those outside of our field. In the introduction – you talk about these as stages of community development. I would consider using terms that evoke the seed to seedling transition vs. effects on an established community.

c. The same can also be said for abbreviations like ALAN but maybe this is a bit nitpicky :)

d. I also found the names of models like "factor identity" to reduce the readability of the text and

make it harder to see the real impact.

2. I think that the introduction is very clearly written but the storyline can be better highlighted especially for a more general audience like that of Nature Communications. For example, in the first paragraph the authors end with multiple global change factors but in the next paragraph they start with individual global change factors. This type of transition makes the introduction very choppy. Some suggestions to improve this in the introduction:

a. Take the second half of paragraph one (starting on line 47) and switch it with the first half of paragraph two (lines 51 to 58). This would give you one paragraph on single factors and their effects and then one paragraph on multiple factors and their interactions.

b. I also think that lines 82 to 85 re: the Komatsu et al. paper should go into this second paragraph since it is not referring to the specific method of Rillig et al. for looking at multiple global change factors but rather that multiple global change factors interacting cannot be predicted by single global change factors in isolation.

c. Revise lines 86 – 88 to talk about how seed germination and establishment may be the most strongly effected and move the point that they receive only minor attention to the end of the paragraph. Right now this is a bit redundant with the need to study this which is obvious from the rest of the paragraph and can be removed.

3. I have some concerns over the model selection procedure. I know that many people use the approach outlined by the authors to narrow down a long list of models and highlight factors that are important. But model selection based on AIC can result in models that include spurious factors because the benefit of including a relevant covariate outweighs the cost of including a spurious one. This is by design with AIC analysis because the goal is to create a predictive model without needing to use an independent dataset for prediction (Treddenick et al. 2021 in Ecology "A practical guide to selecting models for exploration, inference, and prediction in ecology" [Editor's note: the reviewer appended a copy of the paper to their report, see attached PDF]). Based on my understanding of the models in this paper (from Figure 1 and table 1) – the models are nested and it would therefore be more appropriate to use a likelihood ratio test starting from the null model for this kind of inference. I should note – I do not think that this analysis will change the major take home message or novelty of the paper (that individual effects of global change factors are not great predictors of the effect of multiple global change factors) but I do expect that it will change some of the individual results especially with regard to some of the single factor analyses.

4. I feel like the discussion can be bolstered a bit better by existing literature. Papers like those by Komatsu and colleagues (in the introduction) and Rillig et al. (also cited throughout the paper) can really underscore the findings in the paragraph on lines 346 to 356. Further – a lot of the literature from NutNet (especially Harpole et al. 2016, included in this review) will support the increase of productivity and simultaneous decrease in diversity especially because the authors note that they believe there is a strong factor identity effect here.

5. It wasn't clear to me where the assertion that global change may have a strong effect on plant performance during the germination phase at the species level came from. This may be especially difficult to see because the authors do not include biomass for the sown communities at least in table 1.

Minor comments:

Lines 76 – 80: I don't think it's necessary to include the selection effect and the complementarity effect here. These refer to very specific measures from Loreau and Hector 2001 that are not being used here. What the authors are referring to is much more general (best referred to as a sampling effect) and this introduces unnecessary confusion.

Response letter to the reviewers' comments on the manuscript "Multiple simultaneously acting global change factors affect productivity, composition and diversity of grassland-plant communities"

General response to the reviewers' comments: We are thankful for the positive feedback on our study as well as for the useful comments by the reviewers about how the manuscript could be further improved. We carefully revised the manuscript based on these comments and believe that the revised version now addresses all points raised by the reviewers. In brief, next to specific improvements of the main text suggested by the reviewers (e.g. incorporation of a more theoretical basis in the introduction and refining the discussion), we also performed additional analyses to corroborate our main conclusions. Please, see our detailed responses below. Because the additional analyses increased the number of tables and figures in the Supplements, we partly compensated for this by removing the analyses of the diversity measures that were based on the functional group categories.

The line numbers refer to the file "MS_Global_change_and_plant_communities_revised".

Reviewer 1 comments to the authors:

The manuscript presented by Speisser et al presents the results of a pot experiment that investigates the effects of multiple simultaneous global change drivers on plant community composition and productivity. The study builds upon work by Rillig et al to examine not the exact combinations of global change factors manipulated, but rather the number of factors (factor richness). On the whole, I appreciate this approach to addressing the important question of consequences of multiple interacting global change factors. However, I have a few major comments related to the structure of the manuscript and statistical analysis.

1. While it is valid that the number of global change factors examined here cannot be factorially

manipulated, I would suggest that this not form the majority of the set-up of the introduction. Rather, it would be interesting to consider why more simultaneous factors might impact plant communities and growth from an ecological theory standpoint. For example, does it reduce the number of niches available or lead to tipping points? Incorporating a theoretical basis for why the richness of global change factors would matter would be more compelling than the current methods-focused rationale for the experimental design.

Response: We thank the reviewer for this suggestion, and we now incorporated a more theoretical basis and talk about effects of co-acting global change factors on niche diversity by altering resource availability (lines 59 – 68). Furthermore, we introduce that combinations of multiple global change factors could result in synergistic interactions, which could lower critical thresholds, potentially increasing the likelihood that natural systems are pushed beyond tipping points (lines 68 – 75).

2. Similarly, the discussion could benefit from incorporation of more theory. As is, the literature cited is mainly used to either say “they did find the same pattern” or “they didn’t find the same pattern”, with no discussion of why those differences might have been observed.

Response: We now discuss in more detail that the observed effects, and differences to previous findings, are likely brought about by the higher number of combined global change factors. For example, previous studies investigating the combined effects of different global change factors included lower numbers of factors, mainly combinations of two specific factors. Recent studies, however, suggest that factor-richness effects as observed in our study are especially likely to arise by the combination of higher numbers of global change factors (lines 182 – 193).

3. The experimental design to assess community composition needs further clarification. It is unclear why a mixed effects model using scores from the first PCA axis is used to assess community composition, when other statistical techniques are more suited for community

analysis. For example, a PERMANOVA (Anderson, 2001, *Austral Ecology*, 26: 32-46) would be a better choice to assess changes in community composition in the pots and can also be performed using the vegan package in R.

Response: We decided to use the extracted scores of the first PCA axis as response variable in mixed effects models to assess community composition as this allowed us to also incorporate the respective specific factor combinations as random effect in addition to increasing GCF number as fixed effect. Thereby, we are able to account for the fact that there were different specific factor combinations for the respective richness levels. To our knowledge, it is not possible to include such a complex random structure in the PERMANOVA models. Therefore, we would lose important information about the specific factor combinations in such models. However, we acknowledge that PERMANOVA is in principle a good (and standard) way to assess the effects on community composition. Hence, we additionally analyzed the data with PERMANOVA and report the results below. In brief, based on the PERMANOVA models, species composition of both community types was significantly affected by the single factors and by GCF number. These results partly contrast those based on the PC1 scores, where we only found a significant effect of the single factors for the sown communities, and no significant GCF number effects. However, in response to a comment of Reviewer 2, we now also analyzed the effects on community composition based on the PC2 scores and found significant effects of GCF number on the species composition in both community types (Fig. 2; Tables S3, S4). Thus, for the above-mentioned reasons, we keep the PCA-based results in the manuscript. However, if the editor insists, we are willing to also include the PERMANOVA results.

Table A PERMANOVA results for GCF-number and single GCF effects on community composition.

Species Proportions						
Planted Community						
GCF number						
	Df	SumsOfSqs	MeanSqs	F.Model	R ²	p
GCF number	1	0.8092	0.80924	16.3794	0.1002	< 0.001
GCF combination	18	2.3263	0.12924	2.6159	0.28805	< 0.001
Residuals	100	4.9406	0.04941	0.61175		
Single GCFs						
	Df	SumsOfSqs	MeanSqs	F.Model	R ²	p
GCF treatment	6	0.38921	0.064868	1.2555	0.17711	0.044
Residuals	35	1.80837	0.051668	0.82289		
Sown Community						
GCF number						
	Df	SumsOfSqs	MeanSqs	F.Model	R ²	p
GCF number	1	2.0005	2.00046	14.6909	0.09331	< 0.001
GCF combination	18	5.8208	0.32338	2.3748	0.27152	< 0.001
Residuals	100	13.6169	0.13617	0.63517		
Single GCFs						
	Df	SumsOfSqs	MeanSqs	F.Model	R ²	p
GCF treatment	6	1.6775	0.279581	3.1032	0.34725	< 0.001
Residuals	35	3.1533	0.090093	0.65275		

4. Further, Shannon's H index incorporates both evenness and richness into the calculation. Were plants allowed to die in the planted communities in a way that would impact richness? Or was this metric primarily used to assess differential species richness in the sown (seeded) pots? And if so, is there any measurement of germination success in the sown pots to differentiate death vs germination as the source of variation in the response? I would suggest that evenness alone, instead of Shannon's H be used for the planted pots if death was not observed during the course of the experiment (which I think likely given the short-term nature of the study).

Response: Plants in the planted communities were allowed to die, and of the 2160 seedlings planted, 179 seedlings (i.e. 8.3%) died before harvest. Therefore, we think it makes sense to report the diversity for the planted community, but we agree that differences in Shannon's H are mainly due to differences in evenness (lines 217 f., 274 f.). For the sown community, we did not specifically assess species germination but only the number of plants per species at the end of the growth phase, which includes the net effect of both germination and mortality. We agree that separating germination and mortality effects would have been interesting. Unfortunately, however, assessing species germination in the sown community was logistically not feasible. Nevertheless, in the end, the effects on the composition of the sown community remain the same, irrespective of whether they were caused by differences in germination or mortality. However, please, also see our response to the next comment.

5. In the discussion it is implied that the cause was differential germination, but evidence of this should be provided.

Response: Unfortunately, we could not specifically assess the germination rates in the sown community. Still, the final data about species presence also includes effects of different germination rates. In addition, as germination is a major bottleneck, and we know from the data of the planted community that mortality was relatively low, it is likely that the differences visible in the sown community were mainly driven by differences in germination rather than by

differences in mortality. We now included additional information about the species presence in the communities to the supplement. In general, in both community types, the number of individuals per species significantly declined with increasing numbers of co-acting global change factors (Table S7). In addition, however, this negative effect tended to affect more species in the sown than in the planted community, suggesting reduced germination success or increased seedling mortality due to simultaneously acting global change factors (Fig. S3). We now more carefully discuss the cause of the reduced diversity and evenness in the sown community.

6. If assessing the consequences of communities growing from seed vs planted plugs is a primary goal of this study, I would suggest that this should be incorporated into the mixed models as a fixed factor. Otherwise comparisons between the two are difficult to support given the current stats.

Response: In addition to the separate analyses of sown and planted communities, we now performed similar analyses in which we also added the community type and the interaction with the single factor treatments and GCF number (Table S5) and discuss the community type specific responses in the main text (lines 231 – 252). In line with the results of the separate analyses, both community types responded similarly to the individual factors. Also, the responses to the number of factors were largely similar, although the species composition and diversity of the sown communities were generally more strongly affected by simultaneously acting GCFs compared to planted communities (significant and marginally significant GCF number \times Community interactions in Table S5).

7. Clarification throughout of the effects of global change factor richness on the response variables is needed. In the first paragraph of the discussion, the authors suggest that in the planted pots the interaction of multiple factors combined in unique ways that cannot be

predicted by individual factors alone when describing the plant community (lns 277-281), but for the productivity response they suggest that the effects of factor richness might have been driven by the positive effects of eutrophication (i.e., a selection effect) (lns 274-275). Yet the results in Table 1 show for both productivity and community responses, the best model of interaction was “separate-pairwise interactions between factors”. If that were the case, then the same conclusions should be drawn for both community and productivity responses, not the alternative conclusions listed above. However, in lns 336-340 it is stated that the “factor identity model” was the best fit until the separate-pairwise interactions was added to the model. Together the discussion of these factor richness drivers is confusing throughout and care should be taken to unify the discussion text with the statistical results.

Response: We thank the reviewer for pointing this out and carefully revised the discussion to clarify this (e.g. lines 159 – 172, 217 – 230).

Additional minor comments:

8. Somewhere in the title, abstract, and/or keywords, it should be made clear what kind of plant species this experiment was conducted on (i.e., herbaceous species).

Response: We added the specification to the key words and also clarify this in the title by referring to grassland-plant communities (‘Multiple simultaneously acting global change factors affect productivity, composition and diversity of grassland-plant communities’).

9. Some relevant literature is missing from the paper, including the following:

- a. Leuzinger et al, 2011, Trends in Ecology and Evolution, 26: 236-241
- b. Langley et al, 2014, AoB Plants 6: 1-12

Response: We now incorporated the missing literature and thank the reviewer for the suggestions (lines 182 ff., 202 ff.).

10. The distinction between sown and planted communities should be made more clear in the intro/abstract. Different terms might be better to make the distinction (e.g., plugs vs seeded communities).

Response: We thank the reviewer for pointing this out and agree that the difference between both community types might not be obvious to all readers based on the terms ‘planted’ and ‘sown’. However, we think the terms ‘plugs’ or ‘seeded’ do not necessarily make the difference more clear (we were not familiar with the term ‘plugs’ in this context). Thus, while still using ‘planted’ and ‘sown’, we now more strongly emphasize the difference between both community types indicated by these terms, which is the inclusion of additional life stages in the sown community (lines 104 – 107, 111 f., 152 – 155).

11. The entire paragraph on soil biophysical properties in the discussion (lns 323-331) is very speculative given that soils were not measured in this experiment. I would suggest removing this paragraph entirely.

Response: While it is true that we did not perform own measurements, there are previous studies showing clear effects of increasing numbers of global change factors on soil properties (Rillig et al., 2019; Yang et al., 2022). As soil properties play a critical role for plant performance, we still think that it is important to address potential indirect effects via altered soil properties as one potential mechanism of how plants might be affected by simultaneously acting global change factors. However, based on the reviewer’s suggestion, we removed the paragraph but briefly address previous findings (lines 187 – 193).

Reviewer 2 comments to the authors:

The authors conducted a mesocosm study to test the effects of multiple factors on plant biomass and community diversity. The focus is on how the number of global change factors affects plant

communities. The authors conclude that multifactorial combinations yield effects that are not predictable from single factor effects. The topic of how multiple factors interact is of great interest. There have been many individual studies and several reviews focused on this very question with equivocal conclusions.

Response: We thank Reviewer 2 for the critical review, which helped us to further improve our manuscript. In general, while we agree that there are diverse studies and reviews about interactive effects of co-occurring factors, we would like to emphasize two points. First, studies investigating the combined effects of multiple factors so far mainly used combinations of low numbers of factors (mostly two factors). However, there are a few recent studies that found effects of higher GCF numbers on individual plants and on the soil system (e.g., Rillig et al., 2019; Yang et al., 2022; Zandalinas et al., 2021). Second, to the best of our knowledge, our study is the first experimental approach to investigate the effects of a high number of co-acting factors in the context of plant communities, considering, and showing that in addition to individual factor effects also the number of factors itself can affect productivity and composition of plant communities. Please, also see our detailed responses to the specific comments below.

1. The presentation of the data here do not strongly support the conclusions. The simplest explanation for the most notable pattern is that eutrophication, by far the most dominant single treatment, caused the higher biomass in the treatments with higher treatment richness.

Response: We agree that the strong individual effect of eutrophication is also represented in the GCF-number effects, particularly on productivity. However, this does not mean that the observed factor-richness effects could be entirely explained by eutrophication effects. Especially for the factor-richness effects on species diversity and evenness, this is highly unlikely as there was no evidence for any individual factor effects (including that of eutrophication). Also for community productivity, where eutrophication surely is an important

driver of the observed factor-richness effects, the results of the diversity-interaction models indicate that factor interactions contribute to these effects as well. We now emphasize this more strongly in the manuscript (line 157 ff.). Further, we performed additional analyses to disentangle the eutrophication and GCF-number effects (see SI section “*Eutrophication effects*”) and discuss the results in the main text (lines 157 ff., 166 ff.). In brief, the results from these additional analyses support our previous conclusions that while the effects on the productivity of the planted communities can be explained both by GCF number and eutrophication, the effects on species composition and diversity of both community types are due to GCF number itself, independent of eutrophication effects (Table S6).

2. The team used hierarchical diversity models to demonstrate that models with separate pairwise interactions fit the data better than alternatives, indicating that those were better than models with factor identity. I feel like this is a roundabout and opaque way to communicate these data. I would be more satisfied by maybe analyzing a subset of the data with eutrophication removed. That seems like a more straightforward way to eliminate the effects of this dominant treatment. At the very least, there should be some graphical representation of the results that support this primary finding. Can we see the strength or the spread of these two-way interactions?

Response: We now ran additional models to further assess the specific eutrophication effects and report the results (lines 157 – 169; Table S6), which further support our previous conclusions (please, also see our response to the previous comment). Nonetheless, as the hierarchical diversity-interaction modeling framework by Kirwan et al. (2009) was originally developed for estimating contributions of species identities and their interactions to biodiversity effects, we think this is also an appropriate and elegant way to better understand how individual factors but also factor interactions contribute to observed factor-richness effects. Further, we would like to clarify that the aim of the hierarchical diversity-interaction modeling is not to

estimate each individual interaction but to test for the general importance of factor interactions. Below, we added the graphical representations of the two-way interactions for those traits for which the pairwise interactions models had the best fit, showing that, for these models, the interactions generally had more variable, and partly clearly stronger effects than the individual factors.

Figure A Model estimates of the separate-pairwise interaction models for the productivity (a), species diversity (b) and species evenness (c) of the planted community. Explanation of abbreviations: M: ‘initial abundance’ of factors, AF: fungicide, L: light pollution, MP: microplastics, N: eutrophication, S: salinization, IT: increased temperature; asterisks indicate pairwise factor interactions.

3. Additionally, perhaps the average absolute value of treatment effects could be shown instead of means across treatments and treatment combinations that may go in opposite directions and cancel each other out. I need some stronger convincing that eutrophication had no effect on diversity or richness, and that unique combinations of treatments are driving the patterns.

Response: We believe that we already do this, because the treatment effects shown in the figures are from the subset of data on single factors. That is, the black dots in Figs. 1 – 3 represent the effects of the respective individual GCF. In addition, we now added the raw data to these figures to show the spread of the data points. We also did this for the factor-richness parts of the figures, where each data point represents a different factor combination in the factor-richness level. Further, as already mentioned, eutrophication as single factor did not significantly impact species diversity or evenness, neither in the planted nor in the sown community. Therefore, it is unlikely that the highly significant factor-richness effects is driven by the incorporation of an individual factor, which had no significant effect on its own. Our findings that none of the individual factors, including eutrophication, significantly affected species diversity or evenness, and yet there was a highly significant effect of GCF number therefore strongly suggest that the number of co-acting factors itself, rather than the inclusion of any particular factor, had a strong impact on plant community diversity.

4. I'm not sure that PC1 is the best way to capture community differences. I feel like there could be a lot of important info missing in other dimensions.

Response: We agree that PC1 captures only part of the variation and thus that other dimensions might matter as well. Still, similar approaches have been successfully applied before (e.g., Rillig et al., 2019). Moreover, because PC1 explained large proportions of the variation (c. 41% in the PCAs on species proportions; Figure S1), we are convinced that this is a useful and appropriate approach. However, as additional dimension, we now also ran models for PC2, and added the results (lines 127 – 140; Fig. 2; Tables S3, S4). PC1 and PC2 together explained more

than 65% of the variation in the PCAs on species proportions. Based on PC2, we found that GCF number significantly affected the species composition in both community types. In addition, Reviewer 1 suggested to use PERMANOVA to analyze changes in community composition. Although PERMANOVA does not allow us to correctly account for the random structure of our model, we now also did PERMANOVA (see Table A above). These analyses also showed that the individual treatments and GCF number affect community composition.

5. Intro: No rationale is given for hypothesis 3.

Response: We now mention that sown communities could be more responsive than the planted communities because sown communities can be affected already during the critical life-history stages of germination and seedling establishment (lines 91 – 101).

6. Why were these particular treatments chosen?

Response: The individual factors were selected based on the criteria that all of them have become common, widespread environmental changes and are likely to continue to increase in their frequency and intensity, and that they could be easily manipulated experimentally. We now mention this explicitly (lines 302 – 309). Furthermore, we wanted the factors to cover various key properties, such as their physical and chemical nature, as well as their mode of action and their effect directions. Regarding the different factor combinations for GCF-number levels 2 and 4, we did not combine any specific factors on purpose. Instead, to avoid a bias due to uneven representations of factors, we created the combinations randomly with the restriction that each factor was included equally often in each richness level (i.e. each factor was included in two combinations for richness level 2 and in four combinations for richness level 4; lines 320 – 330).

7. Line 108 Do these species co-occur naturally?

Response: According to the FloraWeb database (www.floraweb.de), as well as personal observations, all species co-occur in grassland systems around Konstanz. We now mention that they co-occur naturally (line 286 f.).

8. L137 The supplement says the eutrophication rate was 100 kg ha⁻¹ of Universol® blue (oxide) fertilizer, which included NPK and micronutrients, while the methods say that the eutrophication treatment is 100 kg ha⁻¹ N. NPK would be expected to have much more profound effects than just N bc of nutrient colimitation.

Response: We thank the reviewer for pointing this out. The eutrophication treatment was realized by adding dissolved Universol® blue fertilizer, which for N corresponded to 100 kg ha⁻¹, but was used to simulate general eutrophication instead of pure nitrogen addition. This scenario is more realistic, as most agricultural fertilizers (synthetic fertilizers, manure), but also indirect eutrophication via the leaching of nutrients, do not only increase nitrogen contents. We now clarify this in the supplement and also specify it in the methods of the main text (line 315 f.).

9. L184. Only 2 months may be especially short for diversity patterns to manifest.

Response: We agree that stronger diversity patterns might be visible after a longer experimental period, especially regarding species diversity in terms of the number of individuals per species in the planted community. Nevertheless, our results show that the duration was sufficiently long for clear diversity and evenness patterns to arise in terms of biomass proportions. Indeed, future studies covering several growing seasons to better assess effects on number of species and the number of plants per species will be very interesting, also to see how the biomass-based diversity patterns found in our study will drive species-richness patterns over a longer period (lines 194 – 205).

10. L332. It seems like factor richness has the same effects as we would expect for fertilization: more biomass, less diversity, less evenness. This effect has been shown in many hundreds of studies. I would be surprised if it were not the same here.

Response: We discuss all individual-factor effects on the respective traits (biomass, species diversity, species evenness). Especially for community productivity, we discuss that the strong individual eutrophication effect is also likely to be a driver of the GCF-number effect (e.g. lines 157 ff.). However, the results of the hierarchical diversity-interaction models also indicate that factor identity (including eutrophication) is not the exclusive driver of this pattern, as factor interactions are also likely to contribute to the observed effects. Eutrophication could indeed be an important explanation for the factor-richness effects, especially for productivity of the planted community. However, we did not find effects of any individual factors on species diversity and evenness (Figure 3; Tables S3, S4), and additional analyses showed that GCF-number effects on species composition and diversity were independent of eutrophication effects (lines 166 ff.; Table S6).

11. L340 what is the factor-dominance mechanism? Has that been introduced?

Response: We removed this term to avoid confusion.

12. L349 were there any marginally significant effects? I would guess that eutrophication must have been close. Perhaps, just by chance it did not meet the criteria for significance alone, but in combination with other.

Response: There were no (marginally) significant single-factor effects (including eutrophication) on the species diversity or evenness. Details are provided in Tables S1, S2.

13. I would expect that some of these factors are very trivial for some species (maybe fungicide

has very little influence on a species that relies weakly on mycorrhizal fungi), but eutrophication may have a strong effect on all species.

Response: Indeed, different factors will affect plants to different degrees, depending on the species identity and the treatment factor with its specific properties. So, we agree that, for example, eutrophication can have a stronger direct impact on plants than fungicides. However, we think that this is exactly the reason why effects of global change factors should also be investigated under more realistic conditions, using combinations of global change factors with different properties that co-occur in the environment. The likely variation in species responses to different factors is another reason why we need more multifactor studies. Such studies enable us to draw more general conclusions about how plants and plant communities respond to multifactorial global change, compared to predictions based on the specific effects of certain individual factors, which might in nature be altered by the presence of additional factors. Thus, the aim of our study was to test for effects of increasing numbers of global change factors on plant communities based on a realistic subset of global change factors with different properties (lines 253 – 271).

14. L282. That these factors have different effects on plants

Response: We removed this part from the discussion during the revision, and now focus the discussion more on the effects of simultaneously acting factors, as recommended in other comments of the reviewers.

15. L670 “assess”

Response: Done

16. Throughout: avoid ambiguous antecedents, starting a sentence with “This [verb]...”. It often forces the reader to go backwards before continuing with the sentence.

Response: We thank the reviewer for this advice and we made revisions accordingly.

17. Fig. 2 and 3 are both sandwiched such that the data are difficult to see. If the y-axis were expanded, it would be easier to see the differences between the points and to see the error bars. Also, it would be helpful to see the individual datapoints in each group- it would convey a feel for the replication each mean and error represents.

Response: We thank the reviewer for these suggestions and adjusted the figures accordingly.

Reviewer 3 comments to the authors:

This manuscript by SpeiBer et al. entitled “Multiple simultaneously acting global change factors change plant-community composition and productivity” describes a really lovely test of the hypothesis that simultaneous global change factors may result in different changes to ecosystems than individual global change factors. This paper represents the first effort that I am aware of to test this hypothesis using the number of concurrent global change factors as the main variable in question.

In general – I love this paper. I think it is clearly written and easy to understand and provides great evidence for the importance of thinking about multiple global change factors simultaneously despite the logistical difficulties of doing so. I find the results to be compelling. I also appreciate that the authors take in pieces of the Biodiversity-Ecosystem Functioning framework and use them in an entirely new way that is both unexpected and absolutely spot on. Below – I highlight a few ways in which I think that this already great manuscript can be improved.

Response: We thank Reviewer 3 for this very positive feedback, as well as for the valuable comments to further improve our manuscript.

1. In general, while I found the manuscript to be very clear and largely in active voice. I found

the use of lots of specific terms to be quite jargony. As someone in the BEF field – these terms don't necessarily make it harder for ME to understand the paper but I am concerned that they will decrease the impact of the message of this paper to people outside of this relatively narrow field. Considering the diverse readership of Nature Communications, I would suggest that the authors exchange these terms for phrases that better describe their meaning. Some examples below (though this is a non-exhaustive list just examples that stuck out).

a. Factor richness – I appreciate this analog to species richness but I think it obscures what the variable really means which is just the number of simultaneous global change factors applied. I would use this more intuitive phrase to make it more clear to the readers.

Response: We thank the reviewer for this suggestion and revised the manuscript accordingly, now referring to the number of simultaneously acting global change factors (GCFs) and GCF number.

b. Planted vs. sown communities – I think planted vs. sown is a good distinction for plant ecologists but won't evoke an obvious difference to those outside of our field. In the introduction – you talk about these as stages of community development. I would consider using terms that evoke the seed to seedling transition vs. effects on an established community.

Response: To avoid that the manuscripts get too wordy, we prefer to keep sown vs planted communities, but we now more frequently remind the reader that they differ with regard to the life-history transitions that can be affected by the treatments (e.g. lines 152 – 155). Please, also see our response to comment 10 of Reviewer 1.

c. The same can also be said for abbreviations like ALAN but maybe this is a bit nitpicky :)

Response: We replaced 'ALAN' by the more general and intuitive term 'light pollution'.

d. I also found the names of models like “factor identity” to reduce the readability of the text and make it harder to see the real impact.

Response: We had named the models as in Kirwan et al. (2009), but we agree that those terms are not very clear in the context of our study. We have now renamed the models (1. Null model, 2. GCF-identity model, 3. Separate pairwise GCF-interactions model, 4. Average GCF-interaction model, 5. Additive GCF-specific interactions contributions; e.g. Fig. 4; Tables 1, 2) and we hope that this has improved the readability.

2. I think that the introduction is very clearly written but the storyline can be better highlighted especially for a more general audience like that of Nature Communications. For example, in the first paragraph the authors end with multiple global change factors but in the next paragraph they start with individual global change factors. This type of transition makes the introduction very choppy. Some suggestions to improve this in the introduction:

a. Take the second half of paragraph one (starting on line 47) and switch it with the first half of paragraph two (lines 51 to 58). This would give you one paragraph on single factors and their effects and then one paragraph on multiple factors and their interactions.

b. I also think that lines 82 to 85 re: the Komatsu et al. paper should go into this second paragraph since it is not referring to the specific method of Rillig et al. for looking at multiple global change factors but rather that multiple global change factors interacting cannot be predicted by single global change factors in isolation.

Response: We thank the reviewer for this comment and revised the introduction to have a clearer structure. We now first introduce how individual global change factors can impact ecosystems before explaining the co-occurrence of multiple factors and their interactions, and continue by describing how plant communities could be affected by simultaneously acting global change factors.

c. Revise lines 86 – 88 to talk about how seed germination and establishment may be the most strongly effected and move the point that they receive only minor attention to the end of the paragraph. Right now this is a bit redundant with the need to study this which is obvious from the rest of the paragraph and can be removed.

Response: Done (lines 91 – 101)

3. I have some concerns over the model selection procedure. I know that many people use the approach outlined by the authors to narrow down a long list of models and highlight factors that are important. But model selection based on AIC can result in models that include spurious factors because the benefit of including a relevant covariate outweighs the cost of including a spurious one. This is by design with AIC analysis because the goal is to create a predictive model without needing to use an independent dataset for prediction (Treddenick et al. 2021 in Ecology "A practical guide to selecting models for exploration, inference, and prediction in ecology" [Editor's note: the reviewer appended a copy of the paper to their report, see attached PDF]). Based on my understanding of the models in this paper (from Figure 1 and table 1) – the models are nested and it would therefore be more appropriate to use a likelihood ratio test starting from the null model for this kind of inference. I should note – I do not think that this analysis will change the major take home message or novelty of the paper (that individual effects of global change factors are not great predictors of the effect of multiple global change factors) but I do expect that it will change some of the individual results especially with regard to some of the single factor analyses.

Response: We thank the reviewer for this comment and we now compare the models as recommended (Tables 1, 2). As already predicted by the reviewer, this did not change the outcome.

4. I feel like the discussion can be bolstered a bit better by existing literature. Papers like those by Komatsu and colleagues (in the introduction) and Rillig et al. (also cited throughout the paper) can really underscore the findings in the paragraph on lines 346 to 356. Further – a lot of the literature from NutNet (especially Harpole et al. 2016, included in this review) will support the increase of productivity and simultaneous decrease in diversity especially because the authors note that they believe there is a strong factor identity effect here.

Response: We revised the discussion according to the reviewer’s suggestions (e.g. lines 173 – 179, 182 – 193, 215 – 230).

5. It wasn’t clear to me where the assertion that global change may have a strong effect on plant performance during the germination phase at the species level came from. This may be especially difficult to see because the authors do not include biomass for the sown communities at least in table 1.

Response: We now added more information to show that the sown community was more strongly affected by higher numbers of factors than the planted community was (Figs. S3, S4; Tables S5).

Minor comments:

6. Lines 76 – 80: I don’t think it’s necessary to include the selection effect and the complementarity effect here. These refer to very specific measures from Loreau and Hector 2001 that are not being used here. What the authors are referring to is much more general (best referred to as a sampling effect) and this introduces unnecessary confusion.

Response: We removed the selection and complementary effect to avoid confusion and thank the reviewer for pointing this out.

References

- Kirwan, L., Connolly, J., Finn, J. A., Brophy, C., Lüscher, A., Nyfeler, D., et al. (2009). Diversity–interaction modeling: estimating contributions of species identities and interactions to ecosystem function. *Ecology*, *90*(8), 2032–2038. <https://doi.org/10.1890/08-1684.1>
- Rillig, M. C., Ryo, M., Lehmann, A., Aguilar-Trigueros, C. A., Buchert, S., Wulf, A., et al. (2019). The role of multiple global change factors in driving soil functions and microbial biodiversity. *Science*, *366*(6467), 886–890. <https://doi.org/10.1126/science.aay2832>
- Yang, G., Ryo, M., Roy, J., Lammel, D. R., Ballhausen, M.-B., Jing, X., et al. (2022). Multiple anthropogenic pressures eliminate the effects of soil microbial diversity on ecosystem functions in experimental microcosms. *Nature Communications*, *13*(1), 4260. <https://doi.org/10.1038/s41467-022-31936-7>
- Zandalinas, S. I., Sengupta, S., Fritschi, F. B., Azad, R. K., Nechushtai, R., & Mittler, R. (2021). The impact of multifactorial stress combination on plant growth and survival. *New Phytologist*, *230*(3), 1034–1048. <https://doi.org/10.1111/nph.17232>

Reviewer comments, second round -

Reviewer #1 (Remarks to the Author):

I appreciate the care with which the authors addressed previous reviewer comments. I am largely satisfied with the current submission, aside from one point.

I still find the language of planted vs sown communities confusing. To me, planted is an umbrella term that encompasses both transplanting seedlings and adding seeds. Perhaps "transplanted" vs "seeded" communities would be better terms. Although the definitions as is are reiterated in some places in the manuscript, this terminology is still obscuring the methods. Two reviewers had issues with the terms in the first draft, so I would push back to the authors that this is an important issue to correct, especially because Nature Communications reaches a broad audience.

Reviewer #2 (Remarks to the Author):

The authors performed modified version of a classic biodiversity-ecosystem function study- instead of manipulating diversity and evaluating ecosystem function, they manipulated global change drivers and evaluated ecosystem function, community composition and diversity. I am not impugning the quality of this study or the findings. In fact, I think it is likely true that multiple drivers will reduce diversity below individual drivers. But, the authors make some misleading statements like: "Our study shows that the number of simultaneously acting GCFs can affect the productivity, species composition and diversity of plant communities. " The authors do not clearly address the simplest and most straightforward mechanism to explain a large portion of their results such as GCF richness effects on biomass: a sampling effect. Sampling effect: with higher numbers of drivers (in this study) you are more likely to include the most important driver and therefore, you will see larger effects with higher richness treatment groups (e.g. Wardle, David A. "Is" sampling effect" a problem for experiments investigating biodiversity-ecosystem function relationships?." *Oikos* (1999): 403-407.).

This effect will occur to most readers with some knowledge of the biodiversity- ecosystem function experiments they are mimicking. Many readers know what a sampling effect is, but the phrase is not mentioned. For biomass, Fig. 1 shows exactly what you would expect to see from a sampling effect. Which points in the 2, 4, and 6 groups include eutrophication?

The authors do acknowledge in places, and with other words, that the sampling effect may be present, but in the most prominent sections such as title, abstract, conclusion, they state that the number of factors drivers the patterns. I feel strongly that these should be reworded.

Reviewer #3 (Remarks to the Author):

I am really happy with the revisions to the manuscript and think that the authors have done an excellent job of incorporating my suggestions especially the additional analysis that I suggested in my original review. I think the paper is now stronger and an even better fit for the readership of Nature Communications.

Response letter to the reviewers' comments on the manuscript "Number of simultaneously acting global change factors affects plant species composition, diversity and productivity of grassland communities"

We thank the reviewers for the positive feedback, which has further improved and tailored our manuscript towards publication in Nature Communications.

Please, see our detailed responses to the individual comments below.

Sincerely,

Benedikt Speißer, Rutger Wilschut & Mark van Kleunen

Reviewer #1 (Remarks to the Author):

I appreciate the care with which the authors addressed previous reviewer comments. I am largely satisfied with the current submission, aside from one point.

I still find the language of planted vs sown communities confusing. To me, planted is an umbrella term that encompasses both transplanting seedlings and adding seeds. Perhaps "transplanted" vs "seeded" communities would be better terms. Although the definitions as is are reiterated in some places in the manuscript, this terminology is still obscuring the methods. Two reviewers had issues with the terms in the first draft, so I would push back to the authors that this is an important issue to correct, especially because Nature Communications reaches a broad audience.

Response: We now replaced the term “planted communities”. However, to avoid ambiguity and to point out that just the individual seedlings were transplanted, not an entire existing community, we replaced it by “transplanted-seedling community”, instead of “transplanted community” (e.g. lines 19, 93). In accordance with the author guidance by the editorial office, we kept the term “sown community”. We also asked several native English-speaking colleagues, and they also recommended using “sown community” instead of “seeded community”, as the latter term can also have other meanings.

Reviewer #2 (Remarks to the Author):

The authors performed modified version of a classic biodiversity-ecosystem function study- instead of manipulating diversity and evaluating ecosystem function, they manipulated global change drivers and evaluated ecosystem function, community composition and diversity. I am not impugning the quality of this study or the findings. In fact, I think it is likely true that multiple drivers will reduce diversity below individual drivers. But, the authors make some misleading statements like: “Our study shows that the number of simultaneously acting GCFs can affect the productivity, species composition and diversity of plant communities. “ The authors do not clearly address the simplest and most straightforward mechanism to explain a large portion of their results such as GCF richness effects on biomass: a sampling effect. Sampling effect: with higher numbers of drivers (in this study) you are more likely to include the most important driver and therefore, you will see larger effects with higher richness treatment groups (e.g. Wardle, David A. "Is" sampling effect" a problem for experiments investigating biodiversity-ecosystem function relationships?." *Oikos* (1999): 403-407.).

This effect will occur to most readers with some knowledge of the biodiversity- ecosystem function experiments they are mimicking. Many readers know what a sampling effect is, but the phrase is not mentioned. For biomass, Fig. 1 shows exactly what you would expect to see from a sampling effect. Which points in the 2, 4, and 6 groups include eutrophication?

The authors do acknowledge in places, and with other words, that the sampling effect may be present, but in the most prominent sections such as title, abstract, conclusion, they state that the number of factors drivers the patterns. I feel strongly that these should be reworded.

Response: We appreciate the reviewer's comments, and we now clearly state that the sampling effect can be a contributing factor to the observed patterns. As acknowledged by the reviewer, we already discussed such effects at the relevant passages, especially for the effects on the biomass (e.g. lines 149 f.), but we had not used the term "sampling effect". Using the results of the hierarchical diversity-interaction modelling and the additional analysis of the contribution of the eutrophication treatment (Supplementary Table 6), we now emphasize possible influences of sampling effects in the relevant sections, including the Abstract (e.g. lines 26, 269 f.), Introduction (e.g. line 74), Discussion (e.g. line 151) and Methods (e.g. lines 443, 505). Nevertheless, we would also like to emphasize that the patterns observed for species diversity cannot be explained by the inclusion of a single, strong global change factor (i.e. a sampling effect; Figure 3), suggesting that these effects are driven by interactions among the global change factors, rather than by sampling effects. Regarding the title, we followed the recommendation in the author's guidance by the editorial office, and changed it to "Number of simultaneously acting global change factors affects plant species composition, diversity and productivity of grassland communities". Even though the relationship between productivity and number of simultaneously acting global change factors may be largely due to a sampling effect, we do not believe this changes the fact that we found a

relationship between number of global change factors and productivity. However, if the editorial office wants us to change this, we could remove “productivity” from the title.

Reviewer #3 (Remarks to the Author):

I am really happy with the revisions to the manuscript and think that the authors have done an excellent job of incorporating my suggestions especially the additional analysis that I suggested in my original review. I think the paper is now stronger and an even better fit for the readership of Nature Communications.

Response: We thank the reviewer for this very positive feedback.